# Biological Macromolecule-Based Scaffolds for Urethra Reconstruction

**DOI:** 10.3390/biom13081167

**Published:** 2023-07-26

**Authors:** Saeed Farzamfar, Megan Richer, Mahya Rahmani, Mohammad Naji, Mehdi Aleahmad, Stéphane Chabaud, Stéphane Bolduc

**Affiliations:** 1Centre de Recherche en Organogénèse Expérimentale/LOEX, Regenerative Medicine Division, CHU de Québec-Université Laval Research Center, Quebec, QC G1V 4G2, Canada; saeed.farzamfar@crchudequebec.ulaval.ca (S.F.); megan.richer.1@ulaval.ca (M.R.); stephane.chabaud@crchudequebec.ulaval.ca (S.C.); 2Department of Tissue Engineering and Applied Cell Sciences, School of Advanced Technologies in Medicine, Shahid Beheshti University of Medical Sciences, Tehran 1983963113, Iran; mahya.rahmani@gmail.com; 3Urology and Nephrology Research Center, Shahid Beheshti University of Medical Sciences, Tehran 1983963113, Iran; naji_m_f@yahoo.com; 4Department of Immunology, School of Public Health, Tehran University of Medical Sciences, Tehran 1417613151, Iran; mehdi_aleahmad@live.com; 5Department of Surgery, Faculty of Medicine, Laval University, Quebec, QC G1V 0A6, Canada

**Keywords:** biological macromolecules, urethra reconstruction, tissue engineering, urethra defects

## Abstract

Urethral reconstruction strategies are limited with many associated drawbacks. In this context, the main challenge is the unavailability of a suitable tissue that can endure urine exposure. However, most of the used tissues in clinical practices are non-specialized grafts that finally fail to prevent urine leakage. Tissue engineering has offered novel solutions to address this dilemma. In this technology, scaffolding biomaterials characteristics are of prime importance. Biological macromolecules are naturally derived polymers that have been extensively studied for various tissue engineering applications. This review discusses the recent advances, applications, and challenges of biological macromolecule-based scaffolds in urethral reconstruction.

## 1. Introduction

Urethra tissue engineering (UTE) is a quickly progressing technology that aims to produce an implantable urethral tissue, using a combination of biomaterials, cells, and growth factors. The primary objective of this field is to create a tissue graft that can potentially replace the grafts that are currently used in the clinic [1,2,3,4,5].

Biomaterials are crucial components of UTE as they provide structural integrity for the tissue-engineered scaffolds and support various cellular functions. An ideal scaffolding biomaterial for UTE should be biocompatible, biodegradable, and able to endure mechanical forces [6]. In this context, various types of biomaterials have been explored, including natural polymers, as well as synthetic ones. These biomaterials can be utilized individually or in combination with each other to augment their applicability [6,7]. In particular, these scaffolding systems should be able to promote the growth and differentiation of various cell types that are present in the urethra. Otherwise, the developed tissue’s functionality might be compromised [8,9]. 

Polymers that are derived from large biomolecules, such as proteins, polysaccharides, and nucleic acids, are known as biological macromolecule-based polymers. These polymers consist of repeating monomers linked together by covalent bonds, forming long molecular chains [10,11]. These biomaterials originate from natural sources, and therefore, demonstrate various positive features for interaction with biological systems. Biological macromolecules, such as collagen, chitosan, alginate, cellulose, Hyaluronic acid (HA), gums, and dextran, have been widely explored for different biomedical applications [12,13,14]. Despite being highly biocompatible, scaffolds that are only produced from these polymers have certain shortcomings. In the case of polysaccharides, one of the main challenges is the lack of cell adhesion moieties which can result in reduced tendency of cells towards these polymers [12,13,15]. Furthermore, they may not endure mechanical forces and undergo failure or collapse [16]. Despite these concerns, the biocompatibility, bioactivity, modifiable properties, and versatility of these biomaterials have made them attractive candidates for various tissue engineering applications [10,17]. This review will discuss the previous applications of biological macromolecule-based scaffolds in UTE. In addition, the clinical translation feasibility of these grafts will be discussed, and the current challenges will be highlighted.

## 2. Urethra Defects and Their Current Treatment Options

The urethra is a flexible tubular structure primarily responsible for eliminating urine from the body. In men, the urethra is considerably longer than females (20 cm compared to 4 cm) and is also responsible for the discharge of semen [18]. In males, the urethra consists of the prostatic and membranous segments (posterior urethra) after leaving the bladder. The anterior urethra (bulbar and penile urethra) is the most distal and longest portion, mainly surrounded by the corpus spongiosum. On the other hand, the short female urethra is bordered by the anterior wall of the vagina. Each structural component of the urethra plays a vital and distinct role in its normal function. The malfunctioning of these elements (mucosa, submucosa, and muscle layers) can lead to severe urinary problems [18,19]. The histologic features of the urethral epithelial layer vary, transitioning from a transitional epithelium near the bladder neck to a pseudostratified columnar epithelium in the anterior urethra and a stratified squamous epithelium at the external orifice (meatus). This epithelial layer acts as a highly impermeable barrier against harmful urine elements and protects the submucosa fibroblasts from leakage, preventing inflammatory reactions. The muscular layer is primarily composed of smooth muscle cells arranged in outer circular and inner longitudinal layers. Additionally, striated muscle fibers form a circular arrangement around the membranous urethra in males and the urethra at the pelvic floor level in females, forming a striated muscle sphincter. All these structural components collectively contribute to the passive viscoelastic properties necessary for proper urethral function [19,20].

Any congenital disorder or acquired condition affecting the structural elements of the urethral wall can compromise normal urethral function. Congenital disorders, such as posterior urethral valves, urethral atresia, urethral polyps, hypospadias, epispadias, and megalourethra, can impact normal urethral development and function [21]. Additionally, strictures in the urethral wall caused by traumatic injuries, infection, iatrogenic damage or tumor resection can impair urethral function [22]. Urethral stricture, a relatively common urologic complication affecting approximately 0.6% of men, carries a significant social and economic burden, with reported annual medical costs of around $200 million in 2000 [23]. Stricture formation begins with urethral wall injury, followed by urine extravasation, which triggers a sequence of fibrotic reactions. These reactions alter the normal composition of the urethral connective tissue, resulting in a decrease in collagen type III/I and smooth muscle/collagen ratios (Figure 1) [24,25].

Ultimately, the fibrotic process significantly narrows the urethral lumen, causing symptomatic obstructive voiding. Untreated strictures can lead to complications, such as a thick-walled trabeculated bladder, acute retention, prostatitis, epididymo-orchitis, and hydronephrosis. Consequently, surgical intervention is often necessary in most stricture cases [22].

The choice of surgical method for urethral stricture management depends on the size, location, and recurrence history of the stricture [27]. Urethral dilation, performed through various methods, is suitable for limited and short strictures. Urethrotomy, involving limited incisions in the scar tissue of small strictures to increase the urethral lumen caliber and allow tissue healing, is another method. However, the recurrence rate of urethrotomy increases significantly with the length of the treated strictures [28]. Urethroplasty, with a long-term success rate of 80–90%, is considered the gold standard treatment for urethral stricture [29]. End-to-end anastomotic urethroplasty is an effective method for strictures shorter than 2 cm, but it is not suitable for longer strictures [26]. Urethroplasty using onlay or inlay grafts is the preferred method for extensive strictures due to its straightforward graft harvesting procedure and high long-term success rate. Various graft types from different sources, including skin, bladder mucosa, rectal mucosa, and oral mucosa (buccal or lingual), have been utilized for this purpose [30]. Oral mucosa is the most commonly used graft due to ease of harvest, absence of hair, adaptation to a moist environment, and relatively low morbidity. However, autologous graft application for stricture management has limitations and complications, such as potential donor site morbidity and a shortage of donor tissue for long strictures and repeat procedures [31]. Oral mucosa grafts, in particular, carry risks of donor site contracture, bleeding, infection, numbness, and pain [32]. Tissue engineering of the urethra has emerged as an alternative approach that can address the issues associated with current treatments for urethral stricture disease. Tissue engineering shows promise by providing cell-laden scaffolds and off-the-shelf products for regenerating urethral strictures [33].

## 3. Principles of UTE

There are two primary UTE approaches: acellular scaffolds and cell-seeded scaffolds [34]. These scaffolds come in two main types: synthetic scaffolds made from natural or synthetic biomaterials and biological scaffolds derived from sources such as dermis, small intestine submucosa (SIS), and bladder tissue. Scaffolds intended for UTE should mimic the native urethra’s mechanical properties. The mechanical properties of the urethral tissue show viscoelastic characteristics. These properties remain consistent throughout the tissue without any variations in direction or region. These mechanical properties are closely related to the levels of elastin and collagen present in the tissue [35]. However, replicating this tissue’s mechanical behavior is a challenging task, as the orientation, crosslinking, or composition of the urethra’s ECM affects the overall mechanical strength.

### 3.1. Acellular Scaffolds

Acellular matrices are composed of extra cellular matrix components, which provide the necessary mechanical and biochemical cues for cell adhesion, growth and differentiation. These scaffolds are produced through the removal of all cellular components from a tissue, while leaving the ECM intact with its architecture and composition [36]. In the context of UTE, acellular matrices have been used by different teams to create scaffolds for developing the urethral tissue. Different types of scaffolding systems include SIS, bladder matrix, dermis, and a variety of other tissues [37]. One of the major advantages of using acellular matrices is that they can promote tissue regeneration without eliciting a significant immune response related to the cells. Nevertheless, there has been contradictory studies regarding rejection of acellular scaffolds, as many teams reported the incidence of immunogenic reactions after the implantation of different acellular scaffolds [38,39]. In addition, many teams have faced challenges with acellular tissue-based urethral grafts, mainly fibrosis and strictures. For example, Dorin et al. used acellular matrices of bladder submucosa for treating urethral defects in rabbits. Despite the ingrowth of the urethral cells on the matrices, 75% of the rabbits showed strictures and fibrosis in the graft [40]. In addition, Le Roux et al. used acellular SIS to treat different types of strictures of the male bulbar urethra in nine patients. Within 3 months of surgery, stricture occurred in six of the patients [41]. A similar outcome was observed in a study conducted by Hauser et al., where acellular SIS was employed to treat urethral strictures. Out of the five patients included in the study, four experienced the recurrence of a stricture [42]. 

### 3.2. Cell-Seeded Scaffolds

Due to the demonstrated efficacy of pre-seeded scaffolds to enhance tissue regeneration [43], many research groups currently utilize this strategy to repair damaged tissues [34,44]. Many of the acellular scaffolds described previously can be seeded with cells and then grafted. In this context, De Fillipino et al. conducted a study comparing the effectiveness of unseeded matrices with cell-seeded constructs in terms of healing. They utilized autologous bladder epithelial cells and smooth muscle cells from rabbits to seed tubular matrices made from the acellularized bladder. The rabbits implanted with the cell-seeded matrices exhibited no signs of strictures, whereas those implanted with unseeded scaffolds developed strictures and experienced scaffold collapse [45]. Numerous research teams are currently exploring the application of cell-seeded scaffolds. For example, Fossum et al. employed urothelial cell seeding on SIS to treat six patients diagnosed with severe hypospadias. Their findings demonstrated successful bladder emptying for all patients; however, some of them experienced post-intervention complications such as strictures and fistulas [46]. After 7 years, they were all aesthetically satisfied and had a fully functional bladder [47]. In a more recent study, Barbagli et al. conducted research on the use of oral mucosal cells seeded on a biodegradable membrane, which were subsequently cultured to create a graft. This innovative approach was implemented to treat patients with recurrent strictures. The technique exhibited a success rate of 84%. Notably, one of the key advantages of their method is that it necessitates a smaller tissue biopsy, thereby reducing its invasiveness [48]. However, it is important to note that this approach is not without complications. In a related study, Barghava et al. published their findings on the treatment of urethral stricture in lichen sclerosis patients using decellularized dermis seeded with fibroblasts and keratinocytes derived from human oral mucosa. Following the surgery, the initial integration of the graft was 100%; however, over time, some patients developed fibrosis [49]. Moreover, the application of cell-seeded scaffolds extends beyond biological scaffolds. Currently, numerous synthetic and/or natural polymer-based scaffolds are being investigated for the treatment of urethral injuries. For instance, Raya-Rivera et al. utilized PGA/PLGA scaffolds that were seeded with autologous smooth muscle and epithelial cells to address urethral defects. These scaffolds were implanted in five patients with urethral defects. Remarkably, after six years of implantation, all patients exhibited functional urethras, indicating that synthetic polymers such as PLGA can be successfully assimilated by the human body [50]. It is also possible to combine different types of synthetic materials in order to form a scaffold [51,52,53]. 

### 3.3. A New Approach: The Self-Assembly Method

The self-assembly method relies on the cells’ capability to secrete and organize their ECM in the presence of ascorbic acid. The significant advantage of this approach is that it eliminates the need for exogenous materials in tissue development. Due to this advantage, numerous research teams have endeavored to establish a model for urethral repair using the self-assembly method. Magnan et al. successfully created a tissue-engineered tubular genitourinary graft utilizing dermal fibroblasts. The mechanical properties of their graft were subsequently analyzed, revealing its suturability and even greater resilience compared to the native porcine urethra. However, histological analysis demonstrated that the urothelium was not well-differentiated and exhibited similarities to dermal epithelium [54]. Indeed, it was later on proven that the organ-specific character of the fibroblast cells significantly affects the urothelial cells differentiation process [55]. Following this breakthrough, Caneparo et al. conducted a study aimed at developing a urethral substitute. They employed a combination of vesical fibroblasts and dermal fibroblasts to enhance the mechanical properties of the tissue-engineered grafts. Notably, the substitutes containing at least 10% dermal fibroblast composition exhibited favorable mechanical properties, enabling manipulation by surgeons while maintaining high functionality [56]. The self-assembly method they utilized could be a very interesting method for UTE, as it uses autologous cells and thus reduces the risks of rejection from the patient. The schematic illustration of the self-assembly method is shown in Figure 2. 

## 4. General Characteristics of Biological Macromolecules

In UTE, biological macromolecules have emerged as promising candidates to overcome some limitations of current scaffolding biomaterials. Proteins and polysaccharides, incorporated into the matrix of synthetic polymers, may potentially provide adequate structural integrity to protect the urethral lumen from collapse. Specifically, proteins derived from ECM could successfully imitate the native tissues’ microenvironment and can foster cell adhesion and spreading and function [59,60]. Collagen, as the main constituent of native ECM, plays a fundamental role in maintaining tissue structure and function [61,62]. Polysaccharides, which are composed of long-chain carbohydrates, are characterized by biocompatibility, convenient hydrophilicity, adjustable biodegradability, and abundance [63,64]. Moreover, these polymers can be modified to incorporate functional groups or therapeutic agents and modify their properties for specific regenerative applications [65,66,67]. The main polymeric compounds within this class for UTE include cellulose, chitosan, alginate, dextran, gums, and hyaluronic acid (HA). Further details regarding their general properties and their application in urethral reconstruction are provided below.

### 4.1. Collagen

Collagen, an essential structural polymer found in the ECM of various tissues, exhibits several positive features that make it highly suitable for UTE. Firstly, collagen demonstrates high biocompatibility, meaning it is well-tolerated by living organisms and does not impart adverse tissue reactions. This allows collagen to be perfectly integrated with the native tissues and promote cellular functions [62,68]. Another important characteristic of this polymer is its biodegradability. It can be broken down by enzymatic processes within the body, allowing its remodeling by newly synthesized tissue as the scaffold degrades [6,69,70,71]. This feature improves damaged tissue’s repair, ensuring the successful integration of the scaffold with the native tissue. Furthermore, through interactions with cell surface receptors, collagen can activate intracellular signaling systems that affect various cellular functions [6,71,72].

Collagen’s versatility is another key benefit of this protein. It exists in various types, such as type I, II, III, and more, each with distinct characteristics and distribution patterns. This versatility allows for the selection of specific collagen types based on the target tissue to engineer, enhancing therapeutic outcomes. Moreover, collagen improves the formation of new blood vessels and the infiltration of host cells [6,73].

Collagen can be loaded with various signaling cues for specific tissue engineering applications. For instance, functional groups, growth factors, and bioactive molecules have been embedded into the collagen matrix to accelerate tissue regeneration and promote specific tissue functionalities [6,74,75,76]. Furthermore, collagen is readily available from various natural sources, such as human or animal tissues, and can also be produced through recombinant technologies [6,77]. On the negative side, collagen possess poor mechanical properties, may show batch-to-batch variations in properties, and may potentially elicit immunological reactions [78,79].

### 4.2. Cellulose

Cellulose, a polysaccharide present in plant cell walls or produced by some bacterial species, holds great potential for regenerative purposes. Different types of cellulose, including bacterial cellulose (BC), regenerated cellulose, and nanocellulose, offer distinct properties, benefits, and disadvantages in tissue engineering, and understanding these differences is essential for optimizing their application in tissue engineering [80,81,82].

BC, produced by specific bacteria, possesses various positive features that make it highly suitable for developing tissues in vitro. It features a nanofibrillar microarchitecture with high purity, biocompatibility, and biodegradability. BC-based scaffolds provide excellent mechanical properties, which are crucial for supporting tissue repair. However, the production process can be complex and time-consuming, which limits its widespread use [83,84,85,86].

Regenerated cellulose, derived from sources like wood pulp or cotton, has gained significant attention. It can be processed into various forms such as films, membranes, or fibers, providing a versatile platform for various tissue engineering applications. Regenerated cellulose offers good biocompatibility and tailorable properties. However, it has limited biodegradability and can potentially induce long-term foreign body reactions [87,88,89,90,91].

Nanocellulose, comprising nanoscale cellulose fibers or particles, has emerged as a promising material for different biomedical applications. It includes cellulose nanocrystals and cellulose nanofibrils. Nanocellulose provides a high surface area, biocompatibility, and tailorable properties. It can be formulated into hydrogels, aerogels, or composite scaffolds, creating a favorable environment for various cellular activities. Furthermore, nanocellulose can be loaded with bioactive molecules or combined with other polymers to enhance its potential for specific tissue engineering applications. However, challenges persist in terms of large-scale production and standardization of nanocellulose-based materials [92,93,94,95,96]. 

While cellulose-based materials hold immense potential for tissue engineering, they also present certain challenges. One such challenge is the lack of inherent bioactivity and lack of cell recognition sites in their structure. Cellulose-based constructs may require surface modifications or the incorporation of bioactive molecules to improve their regenerative potential. Another challenge is the probability of immune responses or inflammation due to impurities or residual chemicals from processing. Implementing careful purification and sterilization methods is crucial for minimizing these effects. Additionally, the structural rigidity of this polymer may hinder its application in certain tissue engineering scenarios that require flexibility or dynamic mechanical properties [6,97,98,99,100,101].

Despite these challenges, cellulose-based materials offer several advantages in tissue engineering. Firstly, their natural abundance and biocompatibility make them an attractive alternative to synthetic materials. Secondly, cellulose materials can be tailored in terms of porosity, surface topography, and mechanical properties to mimic the ECM of various tissues. This enables improved cellular interactions and tissue integration. Moreover, cellulose can be combined with other polymers, bioactive agents, or growth factors to enhance its functionality and promote specific tissue regeneration processes [6,98,101,102,103,104,105,106].

### 4.3. Chitosan

Chitosan, a natural polysaccharide obtained from chitin, exhibits various positive properties and has gained attention for tissue regeneration. This biological macromolecule possesses excellent biocompatibility, biodegradability, and bioactivity, making it well-tolerated by living tissues. Obtained from sustainable sources such as crustacean exoskeletons and fungal cell walls, chitosan-based scaffolds provide a permissive environment for tissue development. Furthermore, this polymer shows antibacterial properties, potentially reducing the risk of infection in the scaffold’s implantation site. A notable advantage of chitosan is its versatility in being processed into various forms. This property allows us to produce tailorable scaffold design and bioactivity for different applications [6,107,108,109,110,111,112].

Chitosan can be chemically or physically modified to improve its scope of applications. This versatility enables the creation of chitosan-based constructs that can foster tissue regeneration or drive various cellular functions. Chitosan’s positive charge and functional groups facilitate cell attachment. Furthermore, chitosan has the capability to encapsulate and deliver bioactive molecules like growth factors, genes, and drugs, enabling controlled and localized release for tissue regeneration [113,114,115,116,117,118].

Despite its advantages, chitosan does face certain limitations in tissue engineering. One primary challenge is its poor mechanical properties. To enhance the mechanical properties and stability of chitosan scaffolds, reinforcement with filler materials or combination with other materials, such as synthetic polymers, might be pursued [119,120,121,122]. Another factor to consider is the potential of an immunogenic response to chitosan. While it is generally well-tolerated, individual variations in immune reactions may occur [123,124]. Thus, proper purification techniques of chitosan materials are crucial to minimize adverse immune responses or inflammatory reactions. 

### 4.4. Alginate

Alginate, a natural polysaccharide obtained from seaweed, shows distinct characteristics for tissue engineering and drug delivery applications. Its biocompatibility, biodegradability, and gel-forming capabilities have sparked interest to use this biomaterial as a scaffolding system. A notable feature of alginate is its capacity to form hydrogels when exposed to divalent cations, such as calcium ions [125,126,127]. This gelation process occurs quickly under mild conditions, enabling the incorporation of cells and bioactive molecules without compromising their functionality. The resulting gel structure creates a three-dimensional environment that mimics the natural ECM [126,128,129,130]. Additionally, cells tendency towards this polymer is not as strong as collagen [131]. Another significant attribute of alginate is its tunable properties. The gelation process and mechanical strength of alginate hydrogels can be adjusted by modifying factors such as alginate concentration, crosslinking ion type and concentration, and gelation time [128,132,133]. 

Alginate hydrogels also possess a high water retention capacity, creating a hydrated construct that can preserve cellular functions. This property facilitates nutrient exchange, supports cellular functions, and aids in tissue regeneration. Furthermore, the porous nature of alginate hydrogels allows for the diffusion of oxygen, nutrients, and waste products [128,129,134,135].

### 4.5. Hyaluronic Acid (HA)

HA is a naturally occurring polysaccharide found in the ECM of various tissues. This biological macromolecule exhibits biocompatibility, ensuring it is well-tolerated by living tissues without eliciting adverse reactions or inflammation. This quality makes it an ideal choice for scaffolds or matrices employed in tissue engineering [136,137,138,139]. In addition, HA demonstrates exceptional water-retention potential. This capability helps maintain tissue hydration and lubrication, which are critical for optimal tissue function and reducing friction [140,141,142]. The viscoelastic behavior of HA enables HA-based scaffolds to absorb and distribute mechanical forces, mimicking the properties of native tissues [143,144,145]. HA also interacts with receptors present on cell surfaces, such as CD44, regulating cellular activities [139]. HA’s proangiogenic and immunomodulatory properties foster a favorable environment for tissue regeneration [146,147,148,149]. HA can also serve as a carrier for bioactive molecules such as growth factors, peptides, or drugs. Its capacity to retain and release these molecules in a controlled manner can enhance tissue regeneration processes [149]. Despite these positive features, this polymer’s high cost and poor mechanical properties hinder its widespread use in tissue-engineered products [150,151].

### 4.6. Gums

Gums, natural polysaccharides derived from diverse plant sources, have emerged as promising biomaterials for tissue engineering. They demonstrate several advantageous properties that make them appealing for regenerative medicine. However, it is important to acknowledge their limitations. Gums exhibit excellent biocompatibility, ensuring their compatibility with living tissues, and minimizing adverse reactions or toxicity. This characteristic is essential for graft’s integration with the native tissue. Additionally, being derived from plants, these polymers are natural and renewable materials [152,153]. Gums display unique physical characteristics, including customizable viscosity, ease of processing, and gelation, therefore customizable scaffolds can be produced via this polymer [154,155,156]. These scaffolds have demonstrated the ability to absorb significant amounts of water and maintain tissue moisture. Moreover, their surfaces can be easily modified with various bioactive agents to enhance their healing potential [153,157]. However, it is worth noting that gum-based scaffolds generally lack high mechanical properties, making them less suitable for long-term implantation. Additionally, these polymers often lack cell recognition sites, leading to a limited tendency for cells to interact with them [156,158,159,160].

### 4.7. Dextran

Dextran, a type of polysaccharide, has versatile properties and is being extensively explored for tissue engineering applications. Its wide range of properties and biocompatibility make it appealing for UTE. 

This biomaterial exhibits excellent cytocompatibility and imparts no toxicity towards tissues and cells. However, the successful integration of this biomaterial into living tissues requires incorporation of cell recognition peptides such as Arginine–glycine–aspartic acid (RGD) into matrix of dextran-based scaffolds [161,162]. 

One notable characteristic of dextran is its ability to have adjustable physical properties. This adaptability enables the customization of dextran-based materials to suit particular applications. This biomaterial also possesses good water solubility, making it ideal candidate for developing cell delivery systems [163,164,165]. The optimal water retention capacity of these scaffolds establishes a hydrated microenvironment that supports various cellular functions. Presence of various functional groups in the polymeric backbone of dextran enables researchers to surface-modify dextran-based constructs for various tissue engineering and drug delivery applications [166,167,168]. 

## 5. Previous Applications of Biological Macromolecule-Based Scaffolds for Urethral Reconstruction

Although all of the biological macromolecules reviewed in this paper have potential applicability in UTE, the researchers have only focused on collagen, chitosan, BC, and HA. In the following sections, the previous use of these macromolecules in UTE will be discussed.

### 5.1. Collagen-Based Scaffolds for UTE

Collagen scaffolds have long been utilized in various areas of tissue engineering due to their intriguing properties, including biocompatibility with cells and bioactivity. Recently, numerous studies have focused on pre-clinical trials involving animal models to investigate the application of collagen scaffolds for urethral repair. In 2016, Pinnagoda et al. published a noteworthy study on a novel approach to urethral regeneration, employing an engineered acellular collagen scaffold to guide endogenous cell growth. They conducted implantation of these scaffolds in 20 New Zealand white rabbits, and post-operative results revealed that 20% of the rabbits developed both fistulas and stenosis [169]. These complications may arise due to the differential and inadequate integration of acellular scaffolds compared to cell-seeded scaffolds. In an effort to enhance their properties and minimize post-operative complications, other research teams have explored modifications to these collagen scaffolds. One such study was conducted by Jia et al. in 2015, where they employed a collagen matrix modified with collagen-binding VEGF (vascular endothelial growth factor). The hypothesis was that VEGF-modified collagen could significantly enhance vascularization within the neo-urethral tissue. The introduction of this molecule resulted in the promotion of angiogenesis in their experimental model, which was subsequently implanted in a beagle model [170]. The study demonstrated that Collagen-binding vascular endothelial growth factor (CBD-VEGF) had the potential to enhance tissue regeneration and improve the outcomes of urethral reconstruction. However, the reconstructed urethras exhibited structural and functional differences compared to the native ones. Further experiments are necessary to refine the model and optimize it for urethral repair. Another notable study by Nuininga, J.E. et al. focused on a collagen scaffold modified with various factors. They utilized tubular type I collagen supplemented with vascular endothelial growth factor (VEGF) and fibroblast growth factor (FGF-2) to replace a segment of the urethra in a rabbit model. The results indicated that the rabbits achieved normal urination, accompanied by enhanced neovascularization and urothelium formation [171]. These two studies provide evidence that the incorporation of vascularization-associated factors can stimulate angiogenesis, leading to improved success rates of cell-free scaffold grafts by enhancing graft integration. In the context of repairing longer urethral defects, certain research teams have focused on developing tubular scaffolds, which are believed to be more suitable for such cases. One notable study by Orabi et al. involved the seeding of bladder epithelial and smooth muscle cells onto collagen-based tubular matrices. These tubular grafts were then utilized to treat 15 male dogs with extensive urethral defects, resulting in robust tissue development when the collagen scaffolds were seeded with cells [172]. In addition, they conducted an implantation of tubularized collagen scaffolds without cells in six animals, and their findings revealed that non-seeded scaffolds resulted in poor graft integration. The significance of seeding the tubular scaffolds can be attributed to the fact that the seeded epithelial cells establish a barrier that prevents urine leakage and potential fibrosis. Another noteworthy example of utilizing tubularized collagen scaffolds can be seen in a preclinical study conducted by Li et al. In their research, they seeded endothelial progenitor cells and urothelial smooth muscle cells onto a tubular collagen matrix, which was subsequently implanted in New Zealand White rabbits presenting with long urethral defects [173]. The inclusion of multiple cell types in the co-culture approach was intriguing, as previous studies primarily focused on using epithelial cells alone. The incorporation of diverse cell types may contribute to the success of scaffold implantation by establishing a physiological resemblance to native tissue. Both studies highlight the significance of pre-seeding collagen matrices before implantation in animals to prevent strictures and facilitate tissue development. Moreover, collagen is frequently combined with other molecules to create hybrid scaffolds. For instance, Wei et al. fabricated an electrospun scaffold using a combination of PCL (polycaprolactone), silk fibroin, and collagen, which was then seeded with oral mucosal cells for urethral repair [174]. Various tests conducted in this study demonstrated that the scaffold exhibited favorable compatibility with the cells, promoting their growth and proliferation. Although these tests were conducted solely in vitro, this study provided promising insights by illustrating how the different components of the scaffold facilitated cell adhesion and viability. Overall, these findings underscore the importance of utilizing multi-cellular co-culture techniques, pre-seeding strategies, and hybrid scaffold designs to enhance the compatibility and performance of collagen-based scaffolds in urethral repair. Further research and in vivo studies are warranted to validate the potential clinical applications of these approaches. More tests would be needed in vivo to determine whether the hybrid scaffold is suitable for urethral repair in humans. Moreover, hybrid electrospun nanofibers composed of PCL/collagen/silk fibroin and PLA/collagen [175] were introduced as potential candidates for UTE. Versteegden, L.R., et al. published a study in which tubular collagen type I scaffold with noticeable radial elasticity and shape memory effect was constructed by compression of fibrillar collagen. Human epithelial cells successfully adhered to a hollow scaffold under dynamic conditions simulated in a bioreactor to mimic urination conditions [176]. In a similar approach, Chengdan et al. reported the fabrication of PCL/silk fibroin/collagen electrospun fiber loaded with autogenic oral keratinocytes and TGF-β1 siRNA-transfected fibroblasts for urethra reconstruction. The fibrous scaffold provided a convenient substrate for the formation and growth of the stratified epithelial layer and capillary in the rabbit model after six months. While this study showed promising results, other experiments would be required to make sure the RNA interference is free of unintended off-target effects that could impact the different signaling pathways. Another team used collagen in their hybrid scaffold by coating a polyurethane-urea hydrogel with collagen. This scaffold was then seeded with bladder smooth muscle cells to ultimately repair urethral defects in rabbits. The results showed that the cells could grow correctly on the scaffold, as well as the scaffold having good stretching properties and elongation at break. They also reported a significantly low incidence of complications like fistulas and urethral strictures, implying that the use of collagen in their scaffolds helped with the mechanical properties and biological compatibility with the cells used [177]. While their hybrid scaffold showed potential for cell growth, further studies would be needed in order to investigate the long-term effects of the synthetic scaffolds on the human body. Those studies about hybrid scaffolds show that the benefits of each biomaterial can be maximized by combining collagen with them.

### 5.2. Chitosan-Based Scaffolds for UTE

Similar to collagen, chitosan can be utilized in conjunction with other biomaterials to create a sponge-like structure suitable for cell seeding. As an illustration, Magnan et al. successfully seeded urothelial cells, fibroblast cells, smooth muscle cells, and endothelial cells obtained from a porcine bladder biopsy onto a sponge, thus establishing a bladder model. To fabricate their sponge, they combined bovine collagen, chitosan, and chondroitin sulfates, which were dissolved in acetic acid. Similar to the study conducted by Ikeda et al., the solution was then freeze-dried to form the sponge. The outcomes of their investigation demonstrated that the tissue-engineered bladder model achieved sufficient thickness and, significantly, facilitated the formation of tube-like structures resembling capillaries [178]. Although the scaffold in this study did not consist entirely of chitosan, it demonstrated intriguing characteristics associated with the use of this biomaterial. The findings from this research exhibited promising results for bladder tissue engineering applications, and due to the physiological similarities between the bladder and the urethra, chitosan could serve as a potential scaffold for future urethral repair endeavors. While no pre-clinical animal studies utilizing chitosan-based scaffolds for urethral repair have been published thus far, the compelling properties of chitosan make it a viable option for future application in the field of urethral repair.

### 5.3. Cellulose-Based Scaffolds for UTE

BC is a natural polysaccharide that is produced via bacterial fermentation and is highly biocompatible for various biomedical applications. However, BC-based scaffolding systems are bioinert and do not efficiently integrate into the implantation site. Yang et al. modified BC with soy protein isolate (SPI) in order to develop a biomimetic scaffold for UTE [179]. Then, the developed constructs were used to repair a urethral defect in a rabbit model. They showed that urethral defect repair was successful in the rabbits implanted with BC/SPI scaffolds and did not impart any adverse tissue reactions. Despite showing promising results, this study did not compare the healing activity of BC/SPI scaffolds with existing urethral reconstruction materials or techniques. Without such comparisons, it is challenging to assess the relative advantages and disadvantages of BC/SPI compared to other options available in the field. 

Although cellulose-based scaffolds are not biodegraded by human body enzymes and may elicit foreign body reactions, this biomaterial has FDAs approval for wound healing applications. In addition, the structural similarity of nanofibrous BC to native ECM has added to the therapeutic appeal of this polymer. Huang et al. used BC-based porous scaffolds for the delivery of lingual keratinocytes for UTE [180]. They produced BC scaffolds by adding a gelatin sponge into the fermentation process. Then, lingual keratinocytes were isolated from rabbits and seeded onto the porous delivery system. Then, cell-seeded constructs were implanted into a rabbit ventral urethral defect model. After 3 months of follow up, the researchers found that the rabbits treated with lingual keratinocytes-seeded BC scaffolds maintained open the urethra’s lumen and did not show any immunological reactions. However, urethral stricture was found in other groups. Histological evaluations showed that the epithelium in the cell-seeded constructs was intact and continuous during the first month post implantation. However, on the third month, similar epithelium regeneration was seen in 3D porous BC-only scaffolds and BC scaffolds seeded with lingual keratinocytes. This study used lingual keratinocytes as the seeded cells for UTE. While they may demonstrate promising results in the experimental setting, the feasibility, availability, and ethical considerations of using lingual keratinocytes in clinical applications need to be carefully investigated.

BC-only scaffolds do not possess sufficient bioactivity to foster a robust tissue regeneration. Therefore, it is necessary to modify their structure with supporting cells or therapeutic agents. Adipose-derived stem cells (ASCs) are an easily accessible source of stem cells and have a strong secretion profile that can promote the healing of various tissue injuries. In addition, FGFR2 plays a significant role in urinary tract development and repair. Modifying ASCs to overexpress FGFR2 can potentially enhance their secretory function and improve their healing function in urethral defects. In this regard, Zhu et al. used a double-modified sulfated BC for delivering FGFR2-overexpressing ASCs into a New Zealand rabbit model of urethral defect [181]. In vitro results revealed that the modified constructs were conductive for the cells adhesion and proliferation. In addition, an in vivo study showed that the cell delivery system augmented the urethral defects repair by upregulating the Vascular Endothelial Growth Factor A expression. The research presents a novel approach combining modified ASCs and biomaterials for UTE. However, standardization of the fabrication process, quality control, and regulatory approval would be necessary before this approach can be used in clinical practice. Meeting the rigorous standards set by regulatory agencies particularly for using modified stem cells on human subjects can be very challenging. 

Limited angiogenesis and lack of efficient epithelialization hinder satisfactory regeneration of the urethra following injury. Fortunately, due to versatile properties of bacterial cellulose, it can be combined with other scaffolding systems to augment the vascularization process. In this regard, Wang et al. combined bacterial cellulose with bladder acellular matrix in order to develop a biomimetic tissue-engineered construct [182]. The developed constructs were highly similar to native urethral matrix in terms of 3D structure and the presence of necessary ECM components such as collagen, glycosaminoglycans, and pro-angiogenic growth factors. In vitro angiogenesis assay confirmed the formation of capillary-like tubes on the developed scaffolds. The healing activity of these scaffolds was investigated in a rabbit model of urethral defect. Results showed that the grafts were well-tolerated, and they augmented the urethral defects regeneration and angiogenesis. Although this research emphasized the effects of VEGF on the urethra defect repair, the natural healing response in the body is mediated by complex cross-talk between various growth factors, signaling molecules, and cells. Therefore, the effects of other signaling pathways on the urethral defect repair should be investigated.

The bacteria responsible for producing bacterial cellulose such as Gluconacetobacter xylinus can produce this polymer on the surface of other scaffolds. In this regard, a freeze-dried silk fibroin matrix was utilized as a template for developing bacterial cellulose/silk fibroin hybrid scaffold for UTE [183]. Lingual keratinocytes and lingual muscle cells were seeded onto the constructs, and their behavior was investigated. The study showed that the hybrid scaffold provided a permissive environment for cell adhesion and proliferation. The healing efficacy of the developed constructs was investigated in a canine model of urethral defect. After three months of implantation, the animals treated with the hybrid scaffold seeded with lingual keratinocytes and lingual muscle cells significantly alleviated the urethral stricture symptoms and showed no adverse tissue reactions. Although silk fibroin is generally considered compatible with the body, it can still elicit an immune response or inflammatory reactions in certain individuals. This immune reaction has the potential to hinder the integration of tissue or lead to complications in the implanted urethral construct. In a similar approach, Lv et al. developed a composite sponge from combination of gelatin and bacterial cellulose for UTE [184]. The autologous lingual keratinocytes and smooth muscle cells were seeded onto these composite scaffolds and the healing potential of the cell-scaffold constructs was investigated in a canine model of urethral defect. The study showed that the composite scaffolds seeded with both cell types resulted in significant reduction in urethral stricture and a well-organized urothelium. However, it should be noted that the urethral grafts developed from gelatin might not exhibit ideal durability, as gelatin is highly prone to biodegradation.

### 5.4. Alginate-Based Scaffolds for UTE 

Hydrogels can be designed to transport medications or therapeutic substances directly to the affected area of the stricture. These incorporated drugs can specifically target inflammation, stimulate tissue regeneration, or minimize scarring. Consequently, hydrogels contribute to symptom relief and promote the recovery process after urethral injuries. In addition, hydrogels offer lubrication and decrease friction within the urethra, providing relief from discomfort and pain caused by urethral strictures. By establishing a smoother surface, hydrogels facilitate the flow of urine, thereby diminishing the likelihood of additional injury or irritation. In this regard, Kurowiak et al. investigated the in vitro biodegradation of sodium alginate-based hydrogels in a urine environment to study the hydrogels’ potential applicability to treat urethral stricture [185]. They used two different cross-linking agents to develop the hydrogel system. This approach resulted in the production of hydrogels with a range of mechanical properties and biodegradation rate. The mild cross-linking process of calcium alginate hydrogels allows the incorporation of various therapeutic agents without compromising their biological activity. 

Metal stents are commonly used in the clinic for treating urethral stenosis. However, insertion of these tubes into the urethra may cause pain during erection. In addition, the stent may dislocate after the implantation or simply bend or undergo fracture. Therefore, the use of biodegradable polymeric stents may potentially address these issues. In this context, Klekiel et al. produced a biodegradable stent from sodium alginate using a simple cross-linking method [186]. They then implanted the stents into rabbit’s urethra. They showed that their developed stent expanded well inside the urethra and caused no significant damage to the sounding tissues. In addition, the stent allowed free urine flow and met the required mechanical properties for a stent material. Indeed, the incorporation of anti-fibrosis drugs into the matrix of these stents may further enhance their ability to treat urethral stricture.

### 5.5. HA-Based Scaffolds for UTE

Direct endoscopic internal urethrotomy (DIU) has been widely explored for the treatment of urethral strictures. However, due to the high post-operative recurrence rate, the success rate of this treatment strategy to remain low. In a randomized controlled clinical trial, Chung et al. investigated the effects of HA and carboxymethylcellulose injection in inhibiting stricture recurrence after DIU [187]. In this study, 120 patients were recruited, and they were randomly divided into two groups (60 individuals in each group). In the first group, the patients received HA and carboxymethylcellulose instillation following the DIU. In the second group, the patients were treated with lubricant instillation after the surgery. Results showed that visual analogue scale (VAS) pain score and degree of satisfaction were significantly higher in the patients treated with the HA and carboxymethylcellulose injection. In addition, the recurrence of urethral stricture was observed in 5 patients in the HA/carboxymethylcellulose-treated group compared to 11 lubricant-treated patients. Therefore, the local injection of these two biopolymers is effective in preventing urethral structure recurrence after DIU. The inhibitory effects of these two polymers on urethral stricture could be attributed to the lack of cell recognition sites in the structure of these polymers. However, the anti-stricture potential of these polymers might be further enhanced by incorporating anti-stricture drugs into their structure. This can be carried out by physical escalation, surface modification, or incorporation of another drug delivery system such as nanocarriers. 

HA can be used as a bioactivity enhancer of other polymeric scaffolds in UTE. In this regard, Niu et al. surface-coated electrospun silk fibroin scaffolds with HA in order to improve the epithelization process in the urethral tissue reconstruction following injury [188]. Compared with non-modified silk fibroin scaffolds, the constructs that were coated with HA significantly improved the adhesion and proliferation of urothelial cells that were stained positive for uroplakin-3. In vivo study in a rabbit model of urethral injury showed that the silk fibroin scaffolds coated with HA induced the migration of urothelial cells from the adjacent tissues and effectively restored the urothelium barrier. Although surface coating is a simple and straightforward approach for the modification of the polymeric scaffolds, electrospinning the polymeric blending of HA with silk fibroin may provide more stable functionalization. 

Three-dimensional bioprinting technology aims to develop personalized tissues grafts based on patients needs and conditions. In this technology, bioinks are essential as they provide structural support, encapsulate living cells, and enable the incorporation of bioactive factors. They act as a scaffold for tissue formation and help maintain cell viability and functionality throughout the bioprinting process. The composition and properties of bioinks are carefully designed to mimic the native tissue environment and promote successful tissue regeneration. Zhang et al. bioprinted an artificial urethra using a bioink composed of fibrin, gelatin, and HA. The backbone of the scaffolds was fabricated from a polymer blend of PCL and Poly l-lactic acid-co-e-caprolacton [189]. The bioink system allowed for efficient printing of urothelial and smooth muscle cells into the inner and outer layers of the scaffolds. Results showed that the mechanical properties of the developed constructs were comparable with that of native urethras’. In addition, the printed cells preserved their viability even after 7 days of the cells printing and both cell types preserved their specific markers’ expression. While 3D bioprinting has broken new ground in UTE, obtaining the desired mechanical properties, including strength, elasticity, and contractility, remains a difficult task in UTE. Additionally, ensuring proper functionality, such as maintaining urine flow and continence, is a complex endeavor that necessitates additional research efforts. 

## 6. Challenges

UTE has emerged as a promising area of research in the field of reconstructive urology, providing potential solutions for the treatment of urethral defects. In this context, the use of biological macromolecules has gained significant attention in the development of functional and biocompatible urethral tissue substitutes. However, the successful clinical translation of biological macromolecule-based urethral grafts is accompanied by various challenges.

The rapid degradation of biological macromolecule-based scaffolds poses a significant challenge in UTE. While controlled biodegradation is essential for tissue regeneration, excessively fast biodegradation may impede the formation of stable tissue structures and compromise the long-term stability and functionality of the engineered urethral graft [10]. Another challenge is the mechanical weakness exhibited by some biological macromolecules, such as collagen, dextran, alginate, and chitosan, when used as standalone scaffolds. The mechanical properties of the urethral grafts are pivotal for providing structural support and withstanding external forces. Insufficient mechanical properties may result in scaffold collapse and impact tissue remodeling [16,190]. Immunogenicity is also a concern with biological macromolecules. Collagen, for instance, may induce an immune response in certain individuals, leading to inflammation or rejection [62]. Addressing immunogenicity is essential to ensure that adverse tissue reactions will not occur after the graft’s implantation. Although biological macromolecules offer favorable biocompatibility, they may lack the intrinsic bioactivity necessary for stimulating a successful tissue repair following tissue injuries. Without appropriate bioactive cues, the scaffolds may not adequately promote cell attachment (particularly in case of polysaccharide-based scaffolds that lack cell recognition sites), proliferation, and differentiation, thereby limiting their effectiveness in UTE [15,16,191].

Precisely controlling scaffold properties, including porosity, mechanical strength, pore size, and degradation rate, can be challenging with biological macromolecule-based scaffolds, especially because of the batch-to-batch variations in biological macro molecules’ properties [15,192,193]. Tailoring these properties and ensuring the repeatability of the manufacturing process is crucial. In this context, producing these grafts on a large scale while maintaining consistency and quality poses scalability and standardization challenges. The availability of clinical-grade biological macromolecules, high throughput manufacturing technology, and scalability considerations are crucial for successful clinical translation. 

Ensuring proper integration between the engineered urethral tissue and the host tissue is vital for functional outcomes. Biological macromolecule-based scaffolds should facilitate appropriate cell signaling, vascularization, and ECM remodeling to promote seamless integration with the surrounding tissues and restore normal urethral function.

A specific challenge regarding the UTE is developing a functional and continuous urothelium within the engineered tubular tissue. As these scaffolds do not possess a flat surface, establishing a perfect air–liquid interface for the maturation of urothelium is a challenging task. It is crucial to develop methods to generate a urothelial lining that can effectively prevent infections and urine leakage.

Addressing these challenges requires continuous research and development efforts. Strategies such as scaffold hybridization, incorporation of bioactive molecules or growth factors, scaffold surface modifications, vascularization strategies, and advanced fabrication techniques show promise in enhancing the performance of biological macromolecule-based scaffolds for UTE.

## 7. Prospects and Concluding Remarks

Despite the challenges, the use of biological macromolecules in UTE shows significant potential for advancing reconstructive urology and addressing urethral defects and urological disorders. These biomaterials offer distinct advantages that make them promising materials for constructing urethral grafts. Firstly, these macromolecules demonstrate excellent biocompatibility, ensuring minimal adverse reactions when in contact with living tissues. With the advent of new scaffold modification technologies and drug delivery systems, various signaling cues can be incorporated into the matrix of biological macromolecule-based scaffolds that can further enhance their potential for repairing damaged urethra. Furthermore, biological macromolecules offer versatility in scaffold design and fabrication. They can be processed into various forms, such as hydrogels, fibers, membranes, or coatings, allowing for customization according to the specific requirements of UTE. This versatility enables the development of scaffolds with tailored properties.

In conclusion, the use of biological macromolecules in UTE holds great promise. Their biocompatibility, bioactivity, modifiable properties, and versatility make them suitable materials for scaffold construction, leading to potential improvements in treatment outcomes, enhanced tissue regeneration, and restoration of urethral functionality. Continued research and development in this field will contribute to advancing regenerative therapies and benefiting individuals with urethral defects or urological disorders.

## Figures and Tables

**Figure 1 biomolecules-13-01167-f001:**
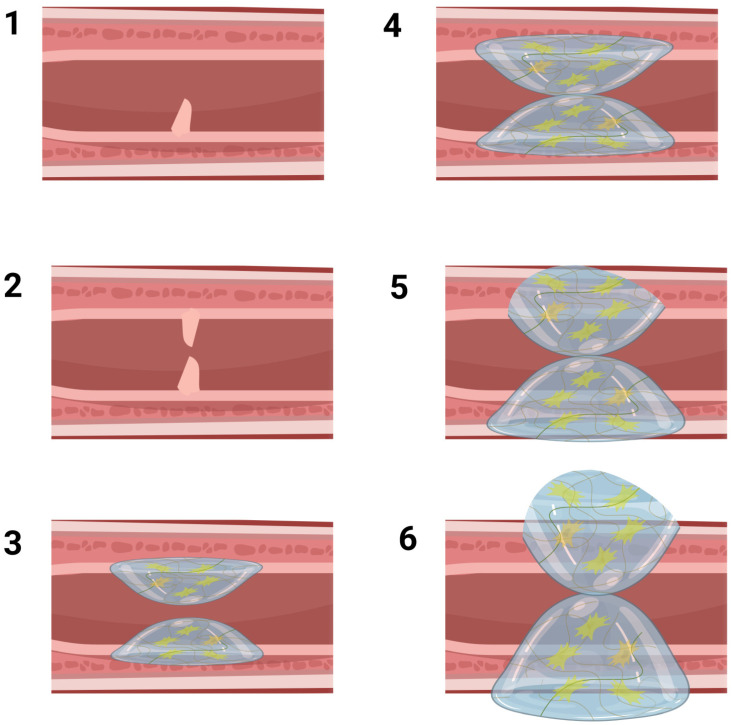
Illustration demonstrating the underlying processes of urethral stricture. Phase 1 displays a folding of the mucosal layer. At this stage, minor abnormalities in the transformed tissue result in the leakage of urine, triggering the initiation of a fibrotic response. Increased accumulation of extracellular matrix (ECM) components leads to the progression into subsequent phases of urethral stricture. Phase 2 depicts constriction resembling an iris. Phase 3 indicates that the fibrotic reaction has infiltrated the spongiosum, causing minimal fibrosis in the sponge-like tissue. Phase 4 represents a partial spongiofibrosis affecting the full thickness of the tissue. Phase 5 demonstrates the extension of fibrosis beyond the corpus spongiosum into surrounding tissues. Phase 6 signifies the development of a complex urethral stricture that may lead to the formation of a fistula. Adopted from reference [26].

**Figure 2 biomolecules-13-01167-f002:**
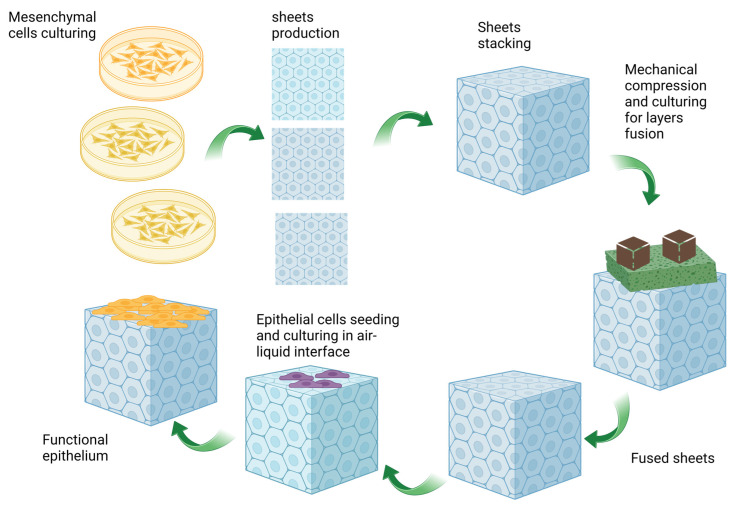
The self-assembly method involves several steps for tissue production. First, mesenchymal cells are cultured in Petri dishes with a paper support and ascorbate (50 µg/mL) for 28 days. This leads to the formation of ECM sheets. These sheets are then superimposed and secured together using surgical clips, and mechanical compression is applied using metal weights (brown cubes). A sponge (the green spongy object) is put between the metal weights and the tissue to protect tissues against mechanical damage. The tissue construct consisting of ECM layers is then cultured to allow fusion between the layers. Next, epithelial cells (purple cells in the figure) are seeded onto the constructs and cultured to populate the surface. Once this is achieved, the cell-scaffold constructs are transferred to an air–liquid interface, which promotes the maturation of the epithelium (yellow cells in the figure). The entire production process takes approximately 60 days. Adopted from references [57,58].

## Data Availability

No data was generated or analyzed in this study.

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
