# Peer review of "Biological Macromolecule-Based Scaffolds for Urethra Reconstruction"

_biomolecules, 2023, doi:10.3390/biom13081167_

Round 1

Reviewer 1 Report

The authors made an analysis of previous studies devoted to using biological macromolecule-based scaffolds for urethra tissue engineering. The review is based on a sufficient amount of actual papers. I would recommend this for publication only after addressing the following issues:

It seems to me, mechanical properties of origin urethral tissue and various scaffolds used for urethra reconstruction are need to be discussed.

Line 135: please, spell out the full term ECM.

Section 3-3: Authors postulate that numerous research teams use the self-assembly method for establishing a model for urethral repair. However, three papers of one research team are cited only. Please, comment it.

Section 4-1 presents only advantages of collagens. Please discuss also their challenges.

Section 4-5 presents only advantages of Hyaluronic Acid. Please discuss also its challenges.

The terms “lichen sclerosis” (Line 177), “tissue-engineered tubular genitourinary graft” (Line 194) “cellulose nanocrystals” (Line 285), and “cellulose nanofibrils” (Line 286) use only one time in the text. Their abbreviations are not needed.

Line 396: please, spell out the full term RGD.

Line 429: please, spell out the full term CBD.

Line 662: please, spell out the full term PLCL.

Author Response

The authors made an analysis of previous studies devoted to using biological macromolecule-based scaffolds for urethra tissue engineering. The review is based on a sufficient amount of actual papers. I would recommend this for publication only after addressing the following issues:

It seems to me, mechanical properties of origin urethral tissue and various scaffolds used for urethra reconstruction are need to be discussed.

Response: the mechanical properties of native urethral tissue was discussed in section 3.

Line 135: please, spell out the full term ECM.

Response: Full term was replaced

Section 3-3: Authors postulate that numerous research teams use the self-assembly method for establishing a model for urethral repair. However, three papers of one research team are cited only. Please, comment it.

Response: the self-assembly method is not the main focus on this paper. Therefore, we did not add many references for this method.

Section 4-1 presents only advantages of collagens. Please discuss also their challenges.

Response: Some challenges with the use of collagen in tissue engineering was discussed in this section.

Section 4-5 presents only advantages of Hyaluronic Acid. Please discuss also its challenges.

Response: Some challenges with the use of HA in tissue engineering was discussed in this section.

The terms “lichen sclerosis” (Line 177), “tissue-engineered tubular genitourinary graft” (Line 194) “cellulose nanocrystals” (Line 285), and “cellulose nanofibrils” (Line 286) use only one time in the text. Their abbreviations are not needed.

Response: abbreviations were deleted.

Line 396: please, spell out the full term RGD.

Response: This term was defined in full

Line 429: please, spell out the full term CBD.

Response: This term was defined in full

Line 662: please, spell out the full term PLCL.

Response: This term was defined in full

Reviewer 2 Report

Dear Author

This review systematically summarizes many papers on research in the field of tissue engineering on urethral reconstruction and will be of great interest to related researchers.

However, there are some details listed below that the authors should address. I hope that my comment is very useful for the improvement of the article.

(1) It is difficult to understand the figures prepared by the authors.

Figure 1.

From the caption, I can imagine that the object in yellow is the "extracellular matrix component," but the authors should clearly illustrate it.

Also, from Phase 5 to 6, this component is popping out of the urethra, but it is difficult to understand what the situation is. The diagram needs to be revisited.

Figure 2.

Please correct the caption to "Figure 1".

Also, please indicate in the legend regarding the yellow-green spongy object, green cubes, red cells, and yellow cells that are present on the cell mass.

(2) In the challenges section, the authors present several challenges with respect to UTE, such as controlling degradability and strength, and suitable integration of the artificial urethral tissue with the host tissue, but these challenges are no different from those of tissue engineering materials in general. Adding a focus on issues in the field of urethral reconstruction would be a better way to raise these issues.

Author Response

This review systematically summarizes many papers on research in the field of tissue engineering on urethral reconstruction and will be of great interest to related researchers.

However, there are some details listed below that the authors should address. I hope that my comment is very useful for the improvement of the article.

(1) It is difficult to understand the figures prepared by the authors.

Figure 1.

From the caption, I can imagine that the object in yellow is the "extracellular matrix component," but the authors should clearly illustrate it.

Also, from Phase 5 to 6, this component is popping out of the urethra, but it is difficult to understand what the situation is. The diagram needs to be revisited.

Response: Figure 1’s caption and its design changed. Phases 5-6 finally leads to formation of a fistula.

Figure 2.

Please correct the caption to "Figure 1".

Also, please indicate in the legend regarding the yellow-green spongy object, green cubes, red cells, and yellow cells that are present on the cell mass.

Response: these objects were explained in the legend.

(2) In the challenges section, the authors present several challenges with respect to UTE, such as controlling degradability and strength, and suitable integration of the artificial urethral tissue with the host tissue, but these challenges are no different from those of tissue engineering materials in general. Adding a focus on issues in the field of urethral reconstruction would be a better way to raise these issues.

Response: Further challenges were added to this section.